# Effect of Psyllium (*Plantago ovata* Forks) Husk on Characteristics, Rheological and Textural Properties of Threadfin Bream Surimi Gel

**DOI:** 10.3390/foods10061181

**Published:** 2021-05-24

**Authors:** Avtar Singh, Soottawat Benjakul, Thummanoon Prodpran, Pornpot Nuthong

**Affiliations:** 1International Center of Excellence in Seafood Science and Innovation (ICE-SSI), Faculty of Agro-Industry, Prince of Songkla University, Hat Yai, Songkhla 90110, Thailand; avtar.s@psu.ac.th (A.S.); thummanoon.p@psu.ac.th (T.P.); 2Office of Scientific Instrument and Testing, Prince of Songkla University, Hat Yai, Songkhla 90110, Thailand; pornpot.n@psu.ac.th

**Keywords:** psyllium husk, threadfin bream surimi, gelation, textural properties, rheology, sensory evaluation, FTIR

## Abstract

Effects of psyllium (*Plantago ovata*) husk powder (PHP) at various concentrations (0, 1, 2, 3 and 4%, *w*/*w*) on gelling properties of surimi from threadfin bream (*Nemipterus* sp.) were investigated. The addition of 1% PHP resulted in the highest increase (50%) in the breaking force (BF) of surimi gel (S), compared to that of the control gel (CON; without PHP). Lower BF was obtained for gel incorporated with PHP at the higher levels (2–4%) (*p* < 0.05). On the other hand, deformation (DF) was decreased with the addition of PHP at all levels compared to the CON gel. The whiteness and expressible moisture content of gels were decreased with augmenting levels of PHP (*p* < 0.05). Protein patterns revealed that PHP at all concentrations did not affect the polymerization of the myosin heavy chain. A loss in the elasticity of the gel was attained with the addition of PHP as indicated by decreased storage modulus (G’). A finer and more compact network was detected in gels containing 1 and 2% PHP than that found in the CON. FTIR spectra suggested that the addition of PHP influenced the secondary structure as well as functional groups of myofibrillar proteins. Based on the sensory evaluation, the surimi containing PHP at 1–3% showed a similar overall likeness score to the CON. Therefore, PHP at the optimum level could improve the gelling properties of the threadfin bream surimi with high acceptability.

## 1. Introduction

Surimi, washed minced fish rich in myofibrillar proteins (MPs), is consumed widely in various forms such as fish balls, imitate crab sticks, and fish tofu due to its distinctive textural properties [1,2]. Usually, lean fish is used for surimi production for whiter products with enhanced gel characteristics than the dark fleshed fish. Threadfin bream is one of the major lean fish used for the surimi production in Thailand [3]. For fish mince, both endogenous transglutaminase (TGase) and proteases are present at various levels depending on species. Heat-induced proteases tightly bind to some fish muscles, which are known to display a damaging impact on the gelling ability of surimi [2]. During the mince-washing process, some endogenous proteolytic enzymes tightly bound to mince cannot be removed, causing gel-weakening (modori) [4]. To tackle this, various additives, such as protease inhibitors [2,5,6], have been employed. Gel properties can be improved by optimizing setting, mediated by endogenous TGase [7]. Calcium ions have been implemented to enhance TGase activity, leading to higher protein cross-linking via non-disulfide covalent bonds [8]. Plant phenolics [9,10] or cross-linking enzymes e.g., microbial TGase [11,12] or chitooligosaccharide [13,14] have been used in surimi. In addition, some hydro-colloids have been employed into surimi to strengthen the gel with high water-holding capacity (WHC) [1,15,16].

Psyllium husk powder (PHP) is a natural source of dietary fiber made from the seed husks of the *Plantago ovata*, an annual plant that belongs to the genus Plantago [17]. PHP is known to contain a high content of water-soluble fiber. It has been used as a prebiotic agent, as well as to control amoebic dysentery, irritable bowel syndrome and intestinal inflammation [18,19,20]. Additionally, PHP showed the ability to reduce lipid and glucose levels [21,22,23]. Moreover, PHP is a highly soluble and viscous fiber, which has been used as a gelling agent. Polysaccharides, such as arabinoxylans (AX), etc. are the major components associated with the high gel-forming property of PHP [18,24,25]. AX is an extremely branched heteroxylan containing 46.8% arabinose and 24.1% xylose, having a 1,4-β-d-Xylo pyranose backbone [26,27]. In general, hydrocolloids, mainly polysaccharides, have been known to influence the gel properties of meat via interaction between protein and polysaccharides. Utilization of PHP in surimi gel could enhance textural characteristics as well as consumer health by lowering bad cholesterol, diabetes and other diseases. Recently, Zhou et al. [28] determined the role of PHP on the gel characteristics of MPs extracted from the pork loin muscle. However, little information is available on the effect of PHP on the gel properties of surimi. Thus, the purpose of the current study was to determine the optimum level of PHP that could be incorporated into the gel without deteriorating the textural properties of threadfin bream surimi. Moreover, consumer acceptability of surimi gel was also elucidated.

## 2. Materials and Methods

### 2.1. Chemicals, Surimi, and Psyllium Husk Powder

All chemicals used were analytical grade and purchased from Sigma (St Louis, MO, USA). Frozen threadfin bream (*Nemipterus* sp.) surimi (grade B) was acquired from Chaichareon Marine Company limited (Pattani, Thailand) and stored at −20 °C for not more than eight weeks. Moisture and protein contents of surimi were 77.60% and 13.23%, respectively. Surimi contained polyphosphate, sorbitol and sucrose at 0.5%, 3%, and 3%, respectively. Psyllium husk powder (PHP) from *Plantago ovata* seed was procured from NOW foods (Bloomingdale, IL, USA). The fiber contents of PHP were determined following the AOAC procedure (985.29) [29]. The total dietary fiber content was 96.01%, in which insoluble and soluble fiber contents were 24.05% and 71.96%, respectively.

### 2.2. Preparation of Surimi Gel

First, the core temperature of frozen surimi was brought to 0–2 °C using tap water, in which it could be cut into pieces with ease. Then, surimi pieces were blended with 2.5% (*w*/*w*) salt using a blender (National (MK-5080M), Selangor, Malaysia) for 1 min to form the paste. Then, PHP was added slowly into the surimi paste (SP) to obtain different final concentrations (0, 1, 2, 3 and 4%; *w*/*w*). The water content of SP was maintained at 80% by adding cold distilled-water, and the mixture was spun for another 2 min. The obtained SP was stuffed into a polyvinylidene chloride casing (diameter of 2.5 cm) and sealed tightly. The temperature of the SP was maintained below 10 °C during the whole preparation. For gelation, two-step heating at 40 and 90 °C was performed for 30 and 20 min, respectively. Successively, all samples were cooled in iced water for 60 min and stored overnight at 4 °C before analyses. The surimi gels (S) containing PHP at 0, 1, 2, 3 and 4% were named as CON, SPH-1, SPH-2, SPH-3 and SPH-4, respectively.

### 2.3. Analyses

#### 2.3.1. Breaking Force (BF) and Deformation (DF)

BF and DF of samples were measured with the help of a TA-XT2 texture analyzer (Stable Micro Systems, Surrey, UK) as per the method of Vate and Benjakul [10]. The spherical plunger (diameter: 5 mm) was used at a compression velocity of 60 mm/min. Before analysis, surimi gels were cut into blocks with a diameter of 2.5 cm with a height of 2.5 cm, which were further conditioned at 25 °C for 1 h.

#### 2.3.2. Expressible Moisture Content (EMC)

EMC of samples was measured following the methods of Buamard and Benjakul [30]. Surimi gels were cut into 5 mm thicknesses and weighed (A). Samples were then put between Whatman paper No. 4 (2 pieces) on the top and three pieces at the bottom of the samples. A standard weight (5 kg) was loaded on the top and left for 2 min. The pressed gel samples were weighed (B). EMC was measured using the following calculation:(1)$$\mathrm{EMC}\;(\%)=\frac{A-B}{A}\;\times \;100$$

#### 2.3.3. Whiteness

Whiteness of samples were measured following the methods of Wijayanti et al. [31]. The color of surimi gel was analyzed using Hunterlab (ColorFlex, Hunter Associates Laboratory, Reston, VA, USA). Lightness (*L**), redness/greenness (*a**) and yellowness/blueness (*b**) were measured by a reflection method. Whiteness index was computed using the equation:Whiteness Index = 100 − [(100 − *L**)^2^ + (*a**)^2^ + (*b**)^2^]^(1/2)^(2)

#### 2.3.4. Protein Pattern

Protein patterns of surimi gels were determined with the aid of sodium dodecyl sulfate-polyacrylamide gel electrophoresis (SDS-PAGE) using the method of Singh et al. [32]. Samples (2 g) were firstly homogenized with 5% (*w*/*v*) hot SDS solution (85 °C) using a homogenizer (IKA Labortechnik, Selangor, Malaysia) at 8000 rpm for 2 min, followed by heating for 1 h at 85 °C. The samples were centrifuged at 7000× *g* for 15 min and subjected to protein determination using the Biuret method [33]. Protein content of each sample was then adjusted to 6 mg/mL and mixed with sample buffer containing 10% (*w*/*v*) β-mercaptoethanol, 4% (*w*/*v*) SDS, 20% (*w*/*v*) glycerol, 0.001% (*w*/*v*) bromophenol blue at a ratio of 1:1 (*v*/*v*) and boiled at 85 °C for 30 min before loading. Samples (15 µg protein) were loaded on 10% (*w*/*v*) running gel and 4% (*w*/*v*) stacking gel and subjected to electrophoresis at a 15 mA current per gel, using a Mini Protein II unit (Bio-Rad Laboratories, Inc., Richmond, CA, USA). Standard markers were also used. After separation, the proteins were stained for 24 h using 0.05% (*w*/*v*) Coomassie Blue R-250 prepared in 50% (*v*/*v*) methanol and 7.5% (*v*/*v*) acetic acid, followed by destaining using the mixture of 30% (*v*/*v*) methanol and 10% (*v*/*v*) acetic acid for 15 min (destaining solution I), followed by 5% methanol (*v*/*v*) and 7.5% (*v*/*v*) acetic acid (destaining solution II) 6 h.

#### 2.3.5. Scanning Electron Microscopic (SEM) Images

Microstructures of samples without and with PHP at various levels were viewed using a field emission-scanning electron microscope (FE-SEM) (Apreo, FEI, Amsterdam, The Netherlands) [12]. For sample preparation, gels with thicknesses of 2–3 mm were fixed with 2.5% (*v*/*v*) glutaraldehyde in 0.2 M phosphate buffer (pH 7.2) for 3 h at room temperature followed by rinsing with distilled water. Fixed specimens were dehydrated in ethanol with serial concentrations of 25, 50, 70, 80, 90 and 100%. Samples were critical point dried using CO_2_ as the transition fluid. The prepared samples were mounted on a bronze stub and sputter-coated with gold and visualized using SEM.

#### 2.3.6. Dynamic Rheology

Surimi pastes added without and with PHP at several levels were made as explained earlier and were analyzed using a HAAKE RheoStress1 rheometer (Thermo Fisher Scientific, Karlsruhe, Germany) following the method of Olatunde et al. [34]. An oscillation frequency of 1 Hz with 1% deformation was used for the analysis. These conditions produced a linear response in the viscoelastic region. The temperature sweep was recorded during heating up from 20 to 90 °C with heating rate of 1 °C/min. Silicone oil was used to minimize water evaporation of surimi pastes during measurement.

#### 2.3.7. Sensory Properties

Sensory analysis of gels added without and with PHP at varying levels was done using a 9-point hedonic scale as per the method of Meilgaard et al. [35] (1, extremely dislike; 2, very much dislike; 3, moderately dislike; 4, slightly dislike; 5, neither like nor dislike; 6, slightly like; 7, moderately like; 8, very much like; 9, extremely like). Eighty panelists (50 male and 30 females with ages of 18–55 years) were recruited for evaluation. Panelists were the students and staffs at the Faculty of Agro-Industry, who had no seafood allergies and often consumed surimi products. A separate sample was served for each panelist. The acceptable limit for the sensory score was 5.

#### 2.3.8. Fourier Transform Infrared (FTIR) Spectra

Spectra of gels were obtained using a Bruker INVENIO S FTIR spectrometer (Bruker Co., Ettlingen, Germany) equipped with an attenuated total reflection (ATR) diamond crystal cell. The samples were freeze-dried using a freeze-dryer (CoolSafe 55, ScanLaf A/S, Lynge, Denmark). The absorption of IR in the region of 400–4000 cm^−1^ was determined using 32 scans with a resolution of 4 cm^−1^. The data were analyzed using OPSU 8.5 software (Copyright © Bruker Optik GmbH 2020, Ettlingen, Germany).

### 2.4. Statistical Analysis

All experiments were conducted in triplicate and a completely randomized design was used. Analysis of variance and comparison of means via Duncan’s multiple range tests [36] was done using an SPSS-23 (SPSS Inc., Chicago, IL, USA).

## 3. Results and Discussion

### 3.1. Change in BF and DF of Surimi Gel Added without and with PHP at Various Levels

BF and DF of all the samples were in the range of 78–118 g and 3.77–5.00 mm, respectively (Figure 1A,B). The CON had the lowest BF (78.25 g), whereas the highest value was obtained for SPH-1 (117.5 g) (*p* < 0.05). The samples containing 2–4% PHP had no difference in BF (*p* > 0.05) but showed a lower BF than that of SPH-1 (*p* < 0.05). With the addition of 1% PHP in surimi, BF was increased by 50% as compared to the CON. Nevertheless, when the concentration of PHP was increased to 2–4%, lower increases (24–28%) in BF were noticed than the CON. The upsurge in BF was most probably related to the gelling ability of polysaccharides (majorly arabinoxylan) present in PHP [27,28]. Polysaccharides can undergo entanglement or interaction with the myofibrillar network, plausibly via H-bonds. This could lead to a strengthened gel matrix. On contrary, Cardoso and Mendes [37] reported a reduction in BF of the gel from unwashed farmed meagre mince prepared using a cold setting with the addition of PHP at both levels (2% and 4%) in combination with 0.5% (*w*/*w*) microbial transglutaminase. PHP rich in polysaccharides with the plenty of OH-groups, which could result in surimi protein and hydro-colloid interactions, mainly via H-bonds. Additionally, PHP contains various kinds of phytochemicals such as alkaloids, flavonoids, tannins, etc. [38]. The presence of OH-groups in these phytochemicals is known to enhance the cross-linking of proteins via hydrophobic interactions, H-bonding, and other interactions [15,39]. The slight decrease in BF with the addition of PHP at higher concentrations was likely caused by the dilution of MPs in surimi. Generally, various interactions with either covalent or noncovalent bonds are formed during the setting phenomenon at 40 °C, in which endogenous TGase plays a major role in the cross-linking of a myosin heavy chain (MHC) via formation of the non-disulfide bond [ε-(γ-glutamyl) lysine linkage], thus strengthening the gel network [13,14] regardless of PHP incorporation. DF was decreased with the addition of PHP, and the lowest value was obtained for SPH-3 and SPH-4 (*p* < 0.05). Nevertheless, similar DF was noted for these two samples (*p* > 0.05). Similarly, CON and SPH-1 had the same DF values (*p* > 0.05). A lower DF value with the addition of PHP was more likely reflected reduction in the elasticity of gel. PHP might impede the interactions between protein chains, especially MPs. Moreover, interaction between proteins via weak bonds could be limited in the presence of PHP, resulting in decreased elasticity of the gels.

### 3.2. Expressible Moisture Content (EMC)

In general, EMC signifies the ability of a gel to absorb water, known as water-holding capacity (WHC). Higher EMC indicates the lower ability of a gel to absorb water [13,14]. During setting, the well-organized arrangement of denatured proteins results in an ordered network with the potential to hold water to a higher extent [40]. Moreover, during heat-induced gelation, cross-linking of denatured and aggregated protein chains can form water channels [41]. Similar EMC was attained between CON and SPH-1 (*p* > 0.05) (Figure 1C). However, with the addition of PHP at higher levels (2–4%), EMC decreased in a dose-dependent manner, and the lowest values were obtained for SPH-3 or SPH-4 (*p* < 0.05). This was plausibly due to the excellent WHC of PHP via the interaction of water with OH-groups of polysaccharides in PHP [42]. BF and DF of farmed meagre surimi gel were deceased with the addition of PHP at 2% and 4%. Nevertheless, WHC was augmented [37]. This reconfirmed the excellent WHC of PHP. During the setting, Zhuang et al. [43] observed the enhanced interactions between proteins via exposed hydrophobic groups, while dietary fiber with abundant hydrophilic groups could form smaller water channels in the gel matrix. This resulted in increased surface area for water molecules to localize in the capillaries of the protein-matrix, which subsequently enhanced WHC [44]. Recently, Zhou et al. [28] observed increasing WHC of the gel prepared from pork loin muscle MPs with the addition of 2% PHP. However, when PHP levels were increased, a lower WHC was noticeable. Thus, PHP at an appropriate level determined the ability of gel network to hold water.

### 3.3. Whiteness

The whiteness of the gel as influenced by the addition of PHP at various concentrations is shown in Figure 1D. The highest whiteness was obtained for the CON (*p* < 0.05). For gels added with PHP, whiteness was reduced with an increasing amount of PHP (*p* < 0.05). This was most likely linked with the brown color of the PHP. Similarly, Zhou et al. [28] documented a reduction in whiteness when PHP concentration exceeded 1%. Cardoso and Mendes [37] observed increasing redness of gel prepared from farmed meagre surimi, with the addition of PHP. Generally, whiteness is related to the type and amount of endogenous pigments in the additives as well as in surimi or fish meat. The addition of certain additives (oils, salts, etc.) can improve the whiteness of gel, especially from dark flesh surimi due to light scattering effect [45]. Therefore, the lower whiteness of gel was governed by the color of PHP.

### 3.4. Protein Patterns

Protein patterns of surimi paste and gels added without and with PHP at various levels are shown in Figure 2. In general, surimi paste comprised myosin heavy chain (MHC) and actin as the foremost MPs, which showed prominent band-intensity. However, upon gelation, the disappearance of the MHC band was noticed in all the samples, which was due to the cross-linking of proteins facilitated by endogenous TGase [1]. Additionally, degradation of MPs during setting mediated by the proteases tightly attached to the muscle, might cause the degradation of MHC [2,4]. MHC has been known to be susceptible to cross-linking during setting. No noticeable differences in actin were obtained in all the samples, irrespective of PHP additions. Generally, actin might not act as a substrate for TGase [46]. When PHP was added at various levels, no change in the MHC band was detected. This suggested that PHP did not interfere with the cross-linking of MHC via non-disulfide covalent bonds. This result reconfirmed that PHP plausibly interacted with MPs by H-bonds and ionic or hydrophobic interactions. These weak bonds could be disrupted in the presence of SDS used during sample preparation [1]. Polyphenols present in PHP, probably at low amounts, did not play a profound role in protein cross-linking. Similarly, another polysaccharide, such as gellan, did not affect the MHC of bigeye snapper surimi at all levels (2–8%) [1]. The results indicate that protein in surimi still underwent cross-linking in the presence of polysaccharides of PHP. However, due to the dilution impact, the strength of protein-protein interactions could be lowered as ascertained by a decrease in BF when PHP at high levels (2–4%) was added.

### 3.5. Microstructure

Microstructures of surimi gel added without and with PHP at varying levels are shown in Figure 3. The CON had a coarser network with larger void or cavities compared with the remaining samples with PHP added. This was in line with the lower BF of the CON (Figure 1A). Samples with PHP added at a lower concentration, such as SPH-1, showed interaction or connections between polysaccharides in PHP and muscle proteins in a good fashion as indicated by a denser and more regular network with smaller and lesser voids or holes. With a further increase in PHP level (SPH-2), more compactness, but greater irregularity were noticed in the protein networks. This indicated that protein strands were filled with the aggregate of PHP, leading to the formation of an interpenetrating gel network. As a result, protein chains interacting with polysaccharides could not aggregate well via several bondings, especially disulfide bonds, which are strong bonds. This led to the lower BF (Figure 1A). Zhou et al. [28] also observed an interpenetrating structure in MP gels prepared from pork loin muscle added with PHP at various levels. A gel network with finer structure and smaller holes can imbibe more water, which agrees with the increasing WHC of gels added with PHP (Figure 1C). Hence, PHP at an appropriate level (1%, *w*/*w*) can improve the three-dimensional network with higher connectivity and order, resulting in enhanced textural properties.

### 3.6. Rheological Properties

Changes in elastic moduli (G′) of surimi gels added without and with PHP at several levels as a function of heating temperature (20–90 °C) are displayed in Figure 4. G′ is a measure of deformation energy stored in the sample during the shear process, which denotes the elastic behavior of a sample [10]. The lowest G′ value was noticed for SPH-4 when heated from 20–38 °C, as shown in the inset. On the other hand, CON, SPH-1 and SPH-2 had similar G′ values in the temperature range of 20–45 °C. With increasing temperature, the CON showed a higher G´ value than the SPH-1 and SPH-2. Conversely, the G′ value of SPH-4 was lower than other samples at temperatures higher than 45 °C. The lowest G′ value for all samples was found in the temperature range of 40–45 °C, which was more likely associated with proteolysis of MPs [2,15]. Generally, this phenomenon was more likely due to the presence of endogenous proteases, which were activated in this temperature range [4]. Additionally, increasing fluidity of surimi paste due to the dissociation, unfolding and denaturation of actin–myosin complex, myosin and meromyosin could be another reason for lower G′ values [47]. With further increases in temperature, a sharp increase in G′ value for all samples was noticed between 60–65 °C. The unfolded proteins were entangled, aggregated, and formed three-dimensional gel networks via the interaction of exposed or unfolded chains [1,30]. However, the highest G′ value was found for CON, followed by SPH-1. It was suggested that the CON could form the aggregation between protein chains via several bonds, especially with strong bonds such as disulfide bonds, to a higher extent in the absence of PHP. This was indicated by the highest G′. For samples with PHP added, the G′ value was decreased with increasing concentration of PHP, especially at high levels (3–4%). In the presence of PHP, the lower interactions between proteins chains could occur to lesser degree, plausibly caused by the interfering effect of polysaccharides in the PHP. This led to the lower G′. When temperature increased to 90 °C, the decrease in G′ was found in all the samples. This might be due to the disruption of weak bonds at higher temperature with high enthalpy. Moreover, this might be due to the rupture of H-bonds during the heating process [48]. A similar trend was reported by Buamard and Benjakul [15] at this temperature range. In this study, the highest BF was noticed for SPH-1 (Figure 1A). In a gel network (solid state), PHP at an appropriate level could enhance the textural properties due to the development of a gel network with interpenetration between MPs and polysaccharides. However, PHP reduced the elasticity of the gel, which was supported by decreasing G′ value with increasing PHP levels. This result was supported by the decreased DF in the presence of PHP (Figure 1B). Therefore, the addition of PHP influenced the viscoelastic property of surimi, which determined the textural characteristics of the resulting surimi gel.

### 3.7. FTIR Spectra

FTIR spectra of SP and surimi gel in the absence and presence of PHP at various levels are shown in Figure 5. For SP, the major peaks characteristics for protein were obtained at wavenumbers of 3281 (*v*_N-H_), 2923 (*v*_C-H_), 1744 (*v*_C=O_), 1642 (*v*_C=O_, amide I), 1538 (*v*_C=O_, amide II), 1451, 1397, 1049, and 991 cm^−1^. The broad absorption at 3281 cm^−1^ (amide A) was derived from stretching of the -NH group, while the band at 2923 cm^−1^ corresponded to C-H stretching [49]. Surimi was added with cryoprotectants such as sugar, which could undergo H-bonding with proteins, resulting in the broad absorption of the NH moiety. The peak at 1744 cm^−1^ found in the SP spectra was mostly due to strengthening of C=O bond in the monosaccharide or sugar added as cryoprotectant [50]. The peak appearing at 1642 cm^−1^ (amide I bond) was assigned to stretching of C=O in the amide of protein. In general, the amide I band representing secondary structure (α-helix, β-sheet, β-turn and random coil structures of proteins) varies between 1600–1700 cm^−1^, respectively [51]. Alternation in the amide I band is generally considered due to the deformation of the α-helix structure to a β-sheet, which has been used to elucidate the gelation mechanism [52]. Before setting (SP), the amide I band appeared at 1642 cm^−1^, which is associated with C=O in an α-helical structure of MPs. Moreover, the amide I band (1642 cm^−1^) is majorly dominated by carbonyl stretching vibrations with minor contributions from C−N stretching and N−H bending, which have been known to change during setting [52]. After heating or setting, the amide I band of SP was shifted to a lower wavenumber of 1639–1634 cm^−1^ (CON and PHP added samples), which was likely due to the formation of β-sheet structures [52]. Also, it indicated the interaction of different functional groups via several bonds induced by heat. Bertram et al. [52] also observed a reduction in amide I wavenumber related to the α-helical structure of MPs during heating. In general, decrease in wavenumber is likely associated with transition dipole coupling when either intramolecular or intermolecular peptides approach each other [52,53]. For the band representing N-H stretching, wavenumber was decreased from 3281 cm^−1^ (SP) to 3276 cm^−1^ (CON). This was likely associated with a participation of this moiety in gel formation. Similarly, amide II (related to -NH deformation) was reduced to 1536 from 1538 cm^−1^ after setting. The bands at 1049 and 991 cm^−1^ belong to C-O stretching, mainly attributed to sucrose present in the surimi [50], which were not affected by setting (SP and CON).

The IR spectra generally showed characteristic absorption peaks of polysaccharides. The IR spectrum is complex, representing a mixture of several polysaccharides present in PHP. Nevertheless, no peak was noticed for amide II. This confirmed the negligible content of proteins in the PHP used in this study. The bands present in high wavenumber regions at 3778, 3322 (broad), and 2922 cm^−1^ are associated with stretching vibrations of free -OH, hydrogen-bonded or clustered -OH and CH/CH_2_ groups, respectively. The bands at wavenumbers of 1635 and 1406 cm^−1^ could arise from bending deformation of -OH and -CH/CH_2_ groups, respectively [54,55]. The bands with wavenumbers in the range of 1200–920 cm^−1^, especially the high intensity peak occurring at 1039 cm^−1^, are generally assigned to the stretching vibration of the C-O bond in ring structures and the C-O-C bond in glycosidic bridges coupled with C-C stretching and C-O-H in-plane bending [55,56].

The presence of peak at 1735 cm^−1^ in the PHP spectrum likely arose from the C=O group of different sugars (mono-, disaccharides) that could be related to the soluble dietary fiber from PHP. When PHP was added into the surimi gel, amplitudes for amide I and II (1538–1535 cm^−1^) bands were reduced, especially at higher concentrations of PHP. This was simply due to the dilution of proteins by PHP addition. Additionally, the wavenumbers of amide A and amide I were decreased to 3277 and 1635 cm^−1^, respectively, with the addition of 3% PHP. This shift to lower wavenumbers indicated the interaction between functional groups of proteins and polysaccharides in PHP, resulting in some changes of secondary structure. The result confirmed the interactions, mostly via H-bonding, between polysaccharides of PHP with surimi proteins, as indicated by alternation in the amide A, amide I and amide II [57]. Moreover, decreased intensity of the amide I and II bands indicated conformational changes occurring during interactions between protein and polysaccharides [57]. With the addition of PHP, the amplitude of bands at wavenumbers of 1049 and 990 cm^−1^ (C-O and C-O-C vibrations) were increased continuously. Sekkal et al. [56] reported that the vibrations of the C-O-C glycosidic linkage in aqueous solutions of oligosaccharides mainly appear in two spectral ranges of 1160–1130 and 999–965 cm^−1^. Therefore, increasing content of glycosidic linkage was associated with a higher concentration of PHP. Those bands slightly shifted to lower wavenumbers, signifying the interactions of oligosaccharides with other components, especially proteins. With the addition of PHP, amide A was broadened, which was associated with increasing stretching vibration modes of -OH groups of polysaccharides present in PHP. It was noted that the amplitude of peaks at wavenumbers of 2914–2923 cm^−1^ were higher in gel with PHP added at a higher level. This reconfirmed the presence of polysaccharides present in PHP, which was added into surimi. The appearance of new bands in surimi gel with PHP added was noticed around 3600–3800 cm^−1^, likely related to the -OH group from phenols (present at lower amount). These interactions were most probably ionic and H-bondings between amino groups (N-H interaction), especially in MP. PHP addition, therefore, altered the secondary structure of proteins, which directly controlled the gelling properties of surimi.

### 3.8. Sensory Property

Likeness scores of threadfin bream surimi gel added without and with PHP at various concentrations are shown in Table 1. For appearance, the highest (7.87) and the lowest (5.80) scores were obtained for CON and SPH-4, respectively (*p* < 0.05). It was noticed that the SPH-1 and SPH-2 shared similar likeness scores (7.13 and 7.20, respectively) for appearance as compared to the CON (*p* > 0.05). SPH-3 (6.47) showed no difference in appearance likeness score with SPH-1 and SPH-2 or SPH-4 (5.80) (*p* > 0.05). The CON, SPH-1 and SPH-2 had the highest likeness scores (7.07–8.00) for color compared to the remaining samples (*p* < 0.05), but no difference in score was noticed among them (*p* > 0.05). SPH-3 and SPH-4 had scores of 5.73 and 5.07, respectively (*p* > 0.05). Although whiteness was decreased with increasing levels of PHP, color acceptability was not negatively affected when PHP was added up to 2% (Figure 1D). There was no difference in likeness scores for odor and taste among all the samples (*p* > 0.05), indicating no offensive odor and flavor associated with PHP. The likeness scores were in the range of 6.53–6.87 and 6.40–7.20 for odor and taste, respectively. For texture, similar likeness scores were noticed for CON, SPH-1, SPH-2 and SPH-3 (6.60–7.27) (*p* > 0.05). On the other hand, SPH-4 (6.07) had the lowest likeness score for texture compared to the other samples (*p* > 0.05). A similar trend was noticed for overall likeness (5.87–7.47). It was noted that the PHP at higher levels had negative impacts on BF or DF, but showed no adverse impact on the consumer acceptability, except for the sample with 4% PHP added, which had a lower acceptability. PHP could be incorporated at high level as an excellent source of dietary fiber since it could lower the risk for developing coronary heart disease, stroke, hypertension, diabetes, obesity, and certain gastrointestinal diseases [58]. Although the likeness scores among the gels sampled with 1–3% PHP added were not different, a level of PHP higher than 1% exhibited a detrimental effect on the textural property of surimi gel, which is its vital property.

## 4. Conclusions

Psyllium husk powder (PHP) directly influenced the gel properties of surimi from threadfin bream. The addition of PHP increased the BF and WHC of the surimi gel. However, it decreased the whiteness as well as viscoelastic characteristics of surimi gel. PHP addition did not affect the polymerization of the MHC. Based on sensory properties, gels added with PHP up to 3% showed no changes in overall-likeness score compared to the CON. A finer and denser network was observed in surimi gel containing 1% PHP. Moreover, PHP could induce an interpenetrating network in surimi gel structure. Thus, PHP at an appropriate level (1%) could be used as an alternative nutritional additive for improving gelling properties, as well as consumers’ health.

## Figures and Tables

**Figure 1 foods-10-01181-f001:**
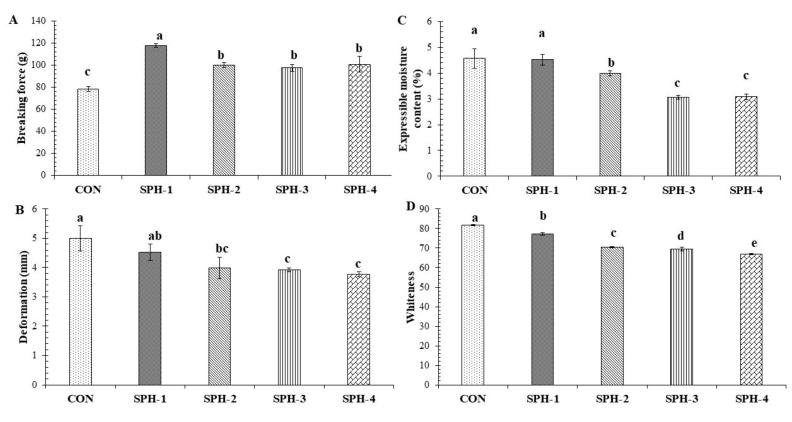
Breaking force (**A**), deformation (**B**), expressible moisture content (**C**), and whiteness (**D**) of gels from threadfin bream surimi without and with psyllium husk at different levels. Bars represent the standard deviation (n = 3). Lowercase letters on the bar indicate significant differences (*p* < 0.05). CON: surimi gel (S) added without psyllium husk powder (PHP), SPH-1, SPH-2, SPH-3, and SPH-4: surimi gel added with 1, 2, 3 and 4% (*w/w*) PHP, respectively.

**Figure 2 foods-10-01181-f002:**
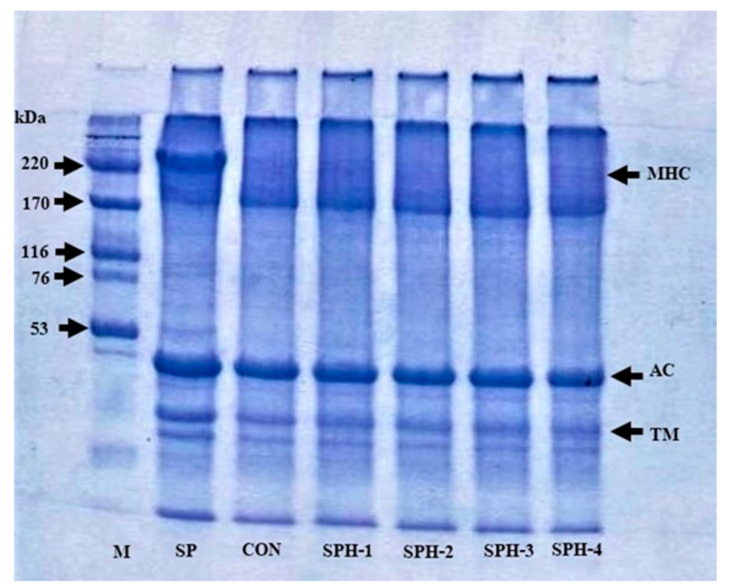
Protein patterns of threadfin bream surimi paste and gel without and with psyllium husk at different levels. M: high molecular weight marker, SP: surimi paste, MHC: myosin heavy chain, AC: actin, and TM: tropomyosin. Caption: See Figure 1.

**Figure 3 foods-10-01181-f003:**
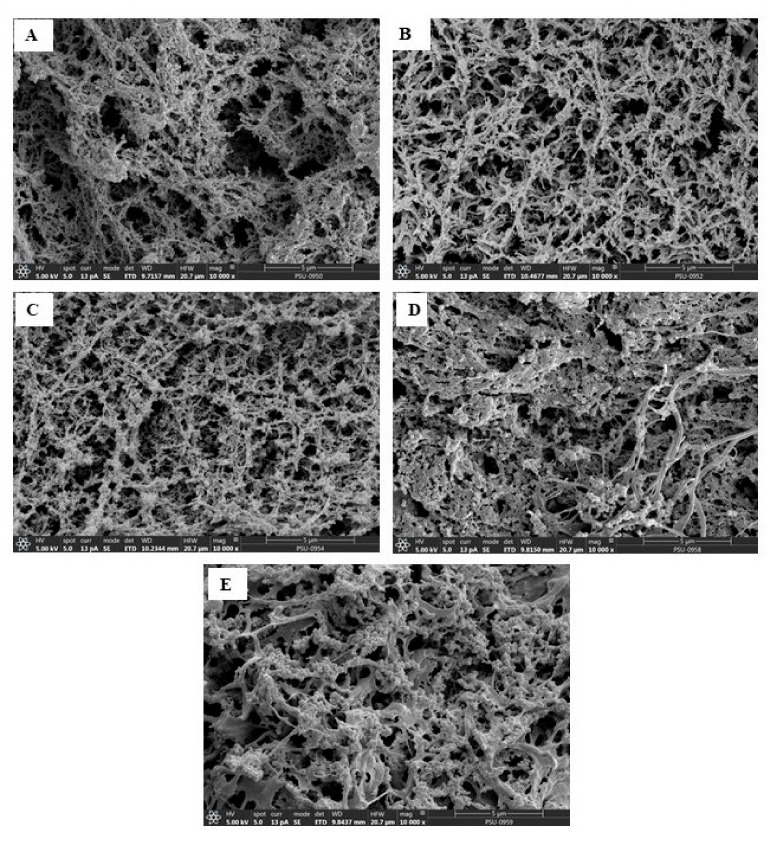
Electron microscopic images of threadfin bream surimi gel without and with psyllium husk at different levels. CON: (**A**), SPH-1 (**B**), SPH-2 (**C**), SPH-3 (**D**), and SPH-4 (**E**). Magnification: 10,000×. Caption: See Figure 1.

**Figure 4 foods-10-01181-f004:**
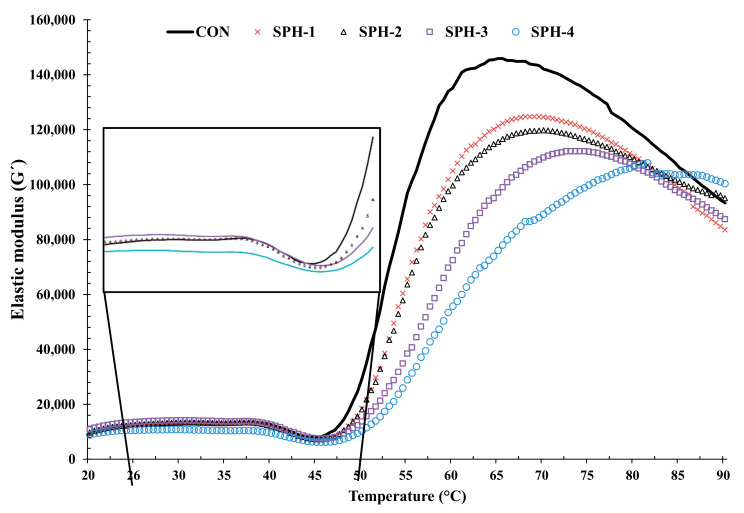
Elastic modulus (G′) during heating (20–90 °C) of threadfin bream surimi paste added without and with psyllium husk at different levels. Caption: See Figure 1.

**Figure 5 foods-10-01181-f005:**
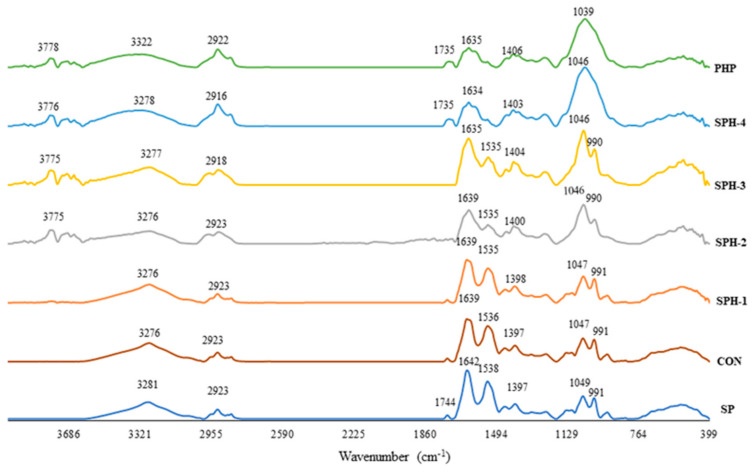
FTIR spectra of threadfin bream surimi gel added without and with psyllium husk at different levels. SP: surimi paste, PHP: psyllium husk powder. Caption: See Figure 1.

**Table 1 foods-10-01181-t001:** Likeness score of thread bream surimi gels added without and with psyllium husk powder at various concentrations.

Samples	Appearance	Color	Odor	Texture	Taste	Overall Likeness
CON	7.9 ± 1.3 ^a^	8.0 ± 1.5 ^a^	6.9 ± 1.6 ^a^	7.3 ± 1.1 ^a^	7.2 ± 0.9 ^a^	7.5 ± 1.0 ^a^
SPH-1	7.1 ± 1.0 ^ab^	7.3 ± 1.1 ^a^	6.8 ± 1.7 ^a^	6.6 ± 1.2 ^ab^	6.9 ± 1.2 ^a^	6.8 ± 1.2 ^a^
SPH-2	7.2 ± 0.9 ^ab^	7.1 ± 1.6 ^a^	6.5 ± 2.1 ^a^	7.1 ± 1.4 ^a^	7.1 ± 1.3 ^a^	7.0 ± 1.3 ^a^
SPH-3	6.5 ± 1.3 ^bc^	5.7 ± 1.6 ^b^	6.6 ± 1.7 ^a^	6.6 ± 1.1 ^ab^	6.4 ± 1.1 ^a^	6.7 ± 1.1 ^a^
SPH-4	5.8 ± 1.3 ^c^	5.1 ± 1.7 ^b^	6.8 ± 1.4 ^a^	6.1 ± 1.2 ^b^	6.4 ± 1.1 ^a^	5.9 ± 1.9 ^b^

Values are mean ± SD (n = 80). The 9-point hedonic scale used, where 1, extremely dislike; 2, very much dislike; 3, moderately dislike; 4, slightly dislike; 5, neither like nor dislike; 6, slightly like; 7, moderately like; 8, very much like; 9, extremely like. Different lowercase superscripts in the same column indicate a significance difference (*p* < 0.05). CON: surimi gel added without psyllium husk powder (PHP), SPH-1: surimi gel (S) added with 1% of PHP, SPH-2: surimi gel (S) added with 2% of PHP, SPH-3: surimi gel (S) added with 3% of PHP, and SPH-4: surimi gel (S) added with 4% of PHP.

## Data Availability

The data are not shared.

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
