# Peer review of "Effect of Psyllium (Plantago ovata Forks) Husk on Characteristics, Rheological and Textural Properties of Threadfin Bream Surimi Gel"

_foods, 2021, doi:10.3390/foods10061181_

Round 1
Reviewer 1 Report
The manuscript includes an interesting and valuable study. I think it need some minor aspects are performed.
Abstract
Lines 12-14: Please, perform what is meant.
Line 20: Based on sensory evaluation, the surimi …
Keywords
Include: sensory evaluation.
Material and methods
Lines 119-122: Include here details provided in the corresponding Table/Figure. Also include information about limiting value, number of panellists, and sharing/not sharing samples by panellists.
Results
According to average sensory values, addition of powder decreased average sensory values. Would it be worth trying with 0.5 % concentration ? Please, include some comment on this, in discussion and discussion.
Author Response
Thank you for the suggestions and valuable time spent on our manuscript. All the queries have been responded and highlighted in yellow.

Reviewer 2 Report
This manuscript deals with the addition of psyllium on rheological and textural properties of threadfin bream surimi gel. This topic has been investigated by Cardoso and Mendes (2013) reference number 34. I don’t see that Your investigation did showed any new findings. If you could investigate the use of psyllium as a cryoprotectant that would be a new finding. The finding of Your study are in some parts contradictory to the study od by Cardoso and Mendes (2013). For example; Line 139 -149. This is very confusing, and needs to explained. You stated that” The samples containing 2-4% PHP have no difference in BrF (p>0.05) but showed a lower BrF than that of SPH-1 (p<0.05). With the addition of 1% PHP in surimi, BrF was increased by 50% as compared to the CON. Nevertheless, when the concentration of PHP was increased to 2-4%, the lower increases (24- 28%) in BrF were noticed as compared to the CON. The upsurge in BrF was most probably related to the gelling ability of polysaccharides (primarily arabinoxylan) present in PHP [27, 28]. Polysaccharides could undergo entanglement or interaction with the myofibrillar network, plausibly via H-bonds. This could lead to a strengthened gel matrix. On contrary, Cardoso and Mendes [34] reported a reduction in BrF of the gel from farmed meagre mince prepared using a cold setting (2 °C) with the addition of PHP at both levels (2 and 4%)”
Your results showed the increase of BrF and the study of Cardoso and Mendes showed the decrease of BrF of surimi with addition of PHP?
Author Response
Thank you for the suggestions and valuable time spent on our manuscript. All the comments are definitely helpful in improving the quality of the manuscript.

Round 2
Reviewer 2 Report
The authors have addressed all points and elaborated all my comments. The paper is now acceptable.